# IMPLICIT INTERMEDIATE SUPERVISION FOR LEARNING COMPLEX FUNCTIONS

## ABSTRACT

Large Language models often rely on explicit intermediate step-by-step supervision, such as chain-of-thought, to solve complex tasks. However, this approach necessitates highly curated data and incurs increased inference time costs. In this study, we investigate the potential of implicit intermediate supervision as an alternative, focusing on multi-task and multi-label learning settings. We demonstrate that training on a dataset with a mixture of tasks allows the learner to utilize the solutions of simpler tasks as intermediate steps for solving more complex ones, reducing the reliance on curated data and explicit supervision. In the multi-label setting, the learner can leverage the signal propagated from easily inferred labels to learn targets that require more subtle computations. We present both theoretical and empirical evidence supporting the notion that neural networks can effectively harness such implicit supervision to tackle complex tasks. Our findings suggest that implicit supervision can shed light on how large language models learn complex tasks while potentially offering valuable insights into developing new versatile methods for solving intricate tasks in language modeling.

## 1 INTRODUCTION

Large Language Models (LLMs), such as GPT-4, Claude-2 and LaMDA (OpenAI, 2023; Anthropic, 2022; Thoppilan et al., 2022), have demonstrated remarkable progress in solving complex multi-step mathematical reasoning and coding tasks (Lewkowycz et al., 2022; Bubeck et al., 2023). This rapid advancement in mathematical reasoning capabilities can be partially attributed to the development of prompting and training techniques that involve intermediate step-by-step supervision, such as Chain-of-Thought (CoT) and scratchpad techniques (Wei et al., 2022; Kojima et al., 2022; Lightman et al., 2023; Nye et al., 2021). In these approaches, when the model is presented with an input question requiring multi-step reasoning, it aims to decompose the problem into multiple steps and solve each step sequentially, rather than directly arriving at the answer. This step-by-step supervision can significantly enhance performance, consistent with theoretical works demonstrating that intermediate supervision is necessary for solving certain problem types (Wies et al., 2023; Malach, 2023).

In this work, we investigate alternative mechanisms that facilitate learning of complex multi-step reasoning without explicit step-by-step supervision. Our primary finding is that neural networks can exploit structure in the training data, which serves as *implicit* intermediate supervision. We examine two mechanisms that allow for such implicit supervision: multi-task and multi-label learning. In this context, we specifically focus on studying feed-forward methods, diverging from auto-regressive models, to explore the potential of learning multi-step reasoning in a single pass through the model, without the multiple iterations and intermediate supervision provided when performing next-token prediction training with auto-regressive inference.

In the multi-task setting, a network is trained on a combination of "easy" and "hard" tasks, where the easy tasks function as sub-tasks for the more complex tasks. For instance, we theoretically investigate the task of *Parity* learning, in which the target is determined by the parity of the sum of an unknown subset of bits in the input (Kearns, 1998; Blum et al., 2003). This task is known to be computationally challenging to learn in various settings, including learning with gradient-descent variants (Shalev-Shwartz et al., 2017; Abbe & Sandon, 2018). We demonstrate that when the network is trained on a multi-task dataset containing both the Parity learning task and the *Sum*

task—where the label is simply determined by the sum of the subset—learning can be accomplished efficiently. We supplement our theoretical findings with experiments, showing that transformers succeed in learning Parities when trained on the Parity-Sum mixture but fail when trained on the Parity task alone. Moreover, we reveal that even if only a small percentage of the data originates from the Sum task, learning is already feasible.

As mentioned, we also study the effect of intermediate supervision in the context of multi-label learning, where a neural network is required to generate multiple predictions for each input. In this setting, our study explores problems characterized by varying levels of difficulty in learning specific labels, with some being relatively easy to learn while others proving more challenging. It is observed that learning the easier labels can facilitate the network's ability to infer the more difficult ones. To investigate this phenomenon, we build upon recent research on synthetic reasoning tasks (Zhang et al., 2023), with a particular focus on code understanding[1]. In this setting, the model is provided with a short synthetic code snippet (see Section 3.3) and tasked with predicting the output generated by executing the given code. For each code example, implicit multi-label supervision is achieved by predicting intermediate steps of the program, such as the stack state. Our observations (see Figure 3) indicate that incorporating this intermediate supervision results in both faster convergence in terms of training steps (or observed samples) and improved final accuracy in predicting the program's output.

The supervision in our setting is considered implicit because the model is not provided with step-by-step guidance or sub-task specific supervision, as in CoT, which is explicit. Instead, the model learns to navigate and solve complex tasks by leveraging the structure and patterns in the mixed task dataset, rendering the supervision implicit.

This study is particularly relevant for understanding LLMs trained on large corpus of data because it explores and analyzes the capabilities and limitations of these models in learning and solving complex tasks without explicit step-wise supervision. In such internet size datasets, implicit supervision can occur unintentionally, for example, comments in coding files provide a signal explaining the current code stack. Thus it offers insights into the mechanisms that enable efficient learning and provides a foundation for improving the training and development of future LLMs.

## 1.1 CONTRIBUTIONS

We summarize our contributions below:

1. We investigate the potential of implicit intermediate supervision in multi-task and multi-label learning settings, demonstrating that neural networks can effectively harness such supervision to tackle complex tasks.

2. We provide theoretical and empirical evidence supporting the notion that training on a dataset with a mixture of tasks allows the learner to utilize the solutions of simpler tasks as intermediate steps for solving more complex ones, reducing the reliance on curated data and explicit supervision.

3. We experimentally show that in the multi-label setting, the learner can leverage the signal propagated from easily inferred labels to learn targets that require more subtle computations, leading to faster convergence and better final accuracy.

4. Our findings contribute to the understanding of how large language models learn complex tasks and offer valuable insights into developing new versatile methods for solving intricate tasks in language modeling.

## 1.2 RELATED WORK

**Mathematical Reasoning with Language Models.** Recent advancements in Large Language Models (LLMs) have demonstrated their potential for solving complex multi-step mathematical and logical reasoning tasks (Lu et al., 2022; Saha et al., 2020; Imani et al., 2023; Kojima et al., 2022). In parallel, LLMs have shown remarkable capabilities in code completion and execution, which often necessitate intricate computations (Chen et al., 2021). However, achieving competitive results

---

[1]Code understanding can be considered a universal reasoning task, as any reasoning chain can be decomposed into code subroutines.

in multi-step reasoning tasks typically requires additional intermediate supervision. For instance, LLMs have been observed to struggle with simple arithmetic tasks when trained without such supervision (Nogueira et al., 2021; Qian et al., 2022). To address this limitation, intermediate supervision can be provided through the Chain-of-Thought method, prompting the model to output intermediate steps before arriving at the final answer (Wei et al., 2022; Lightman et al., 2023). This approach has been shown to enhance model performance in logical reasoning tasks (Kojima et al., 2022; Muffo et al., 2023). Similarly, the "scratchpad" technique enables models to generate intermediate computations arbitrarily when needed for deriving the final answer (Nye et al., 2021). In this work, we investigate the possibility of providing intermediate supervision without explicit guidance via Chain-of-Thought. We focus on the capabilities of neural network models with feed-forward predictions (i.e., without performing multi-step auto-regressive inference) and examine various synthetic mathematical and coding problems involving complex multi-step computations. We demonstrate that training on a mixture of easy and hard problems can enhance performance on such tasks.

**Learning Parities with Deep Networks.** The problem of learning Parities has attracted significant theoretical attention as a canonical hard task for deep learning models (Minsky & Papert, 2017; Barak et al., 2023; Edelman et al., 2023; Telgarsky, 2022; Frei et al., 2023; Shi et al., 2022). This interest is partly due to the fact that learning the Parity (XOR) function is computationally tractable under the Probably Approximately Correct (PAC) framework (Blum et al., 2003), but is hard for many natural learning algorithms (Kearns, 1998). Daniely & Malach (2020) demonstrated that Deep Neural Networks (DNNs) can tractably solve this task when kernel methods fail, given a biased input distribution. Hahn (2019) showed that self-attention does not perform particularly well with parities, while Malach (2023) and Wies et al. (2023) revealed that transformers and even linear auto-regressive models can be improved when Chain-of-Thought is used to provide intermediate supervision. These latter works are particularly relevant to our study, as we demonstrate that this explicit supervision, which also *increases run time computation*, is not necessary when implicit supervision is given.

**Learning from Easy to Hard.** A considerable body of work (Kaiser & Sutskever, 2015; Dehghani et al., 2018; Bengio et al., 2009; Baldock et al., 2021) has been dedicated to understanding how DNNs interact with task complexity. Nakkiran et al. (2019) show that DNNs learn simple functions first and then move on to learn more complex tasks, Schwarzschild et al. (2021); Bansal et al. (2022) show that RNNs can "extrapolate" from easier to harder tasks when given more computation, while some works (e.g., Hacohen & Weinshall (2019) and Gong et al. (2016)) show that a curriculum of increasingly more complex tasks can be beneficial for overall performance. In this work, we demonstrate that complex tasks can benefit from the mere presence of simple tasks in the training distribution, without any added inference time computation.

## 2 THEORY

In this section we give our main theoretical result. We show a simple setting where implicit intermediate supervision allows learning in the multi-task setting. The setting we study has two tasks: Task #1 is the *Sum Task*, where the label is given by the sum of a subset of bits in the input; Task #2 is the *Parity Task*, where the label is given by the parity of the sum computed over the same subset as in Task #1. Our theory shows that, while a neural network fails to learn when trained only on Task #2, training on a dataset containing data from both Task #1 and Task #2 allows the network to succeed on both tasks. Moreover, we show that even a small fraction of data from Task #1 already helps in achieving high accuracy on Task #2.

We start by formally introducing the data distributions generating the data for both tasks. Then, we move on to describing the network architecture and training scheme. Finally, we state the main theoretical results in this setting and discuss the proof.

### 2.1 DATA DISTRIBUTIONS

Let $n$ be some even number. Let $\mathcal{X} = \{\pm 1\}^{n+1}$ be the input space of $n+1$ bits and let $\mathcal{Y} = \mathbb{R}$ be the output space. We define two distributions over labeled examples. For every $\boldsymbol{w}^* \in \{0, 1\}^n$ s.t.

$|\boldsymbol{w}^*| = n/2$ (i.e., exactly half of the coordinates of $\boldsymbol{w}^*$ are 1)[2], we define two distributions, which correspond to the two tasks:

**Sum Distribution (Task #1)**, denoted $\mathcal{D}_{\boldsymbol{w}^*}^{\Sigma}$. Each example $(\boldsymbol{x}, y)$ is drawn as follows:

- The first bit, denoted by $x_0$, is set to 1.
- For each $i = 1, \dots, n$ draw $x_i \sim \text{Bernoulli}(1/2)$, i.e., $x_i = \pm 1$ w.p. $1/2$ each.
- Set $y = \sum_{i=1}^{n} w_i^* x_i - \frac{n}{2}$

**Parity Distribution (Task #2)**, denoted $\mathcal{D}_{\boldsymbol{w}^*}^{\Pi}$. Each example $(\boldsymbol{x}, y)$ is drawn as follows:

- We set $x_0 = -1$.
- For each $i = 1, \dots, n$ draw $x_i \sim \text{Bernoulli}(1/2)$.
- Set $y = \prod_{i=1}^{n} x_i^{w_i^*} = \prod_{i, w_i^* = 1} x_i \in \{\pm 1\}$.

Note that the first bit of the input denotes the task that the learner needs to solve: if $x_0 = 1$ then the output should be the sum, and otherwise the output is the parity. In both cases, the vector $\boldsymbol{w}^*$ denotes which bits of the input participate in the computation of the sum/parity, and these bits are fixed for all examples. Of course, the learner has no knowledge of $\boldsymbol{w}^*$, and needs to find it by learning from the examples.

The setting we analyze is training on a mixture of the two tasks, namely a dataset generated by randomly drawing a Sum example with probability $p$, or otherwise drawing a Parity example with probability $1 - p$. We denote this $p$-mixture of the Sum/Parity distributions by $\mathcal{D}_{\boldsymbol{w}^*}$, where w.p. $p$ we sample $(\boldsymbol{x}, y) \sim \mathcal{D}_{\boldsymbol{w}^*}^{\Sigma}$ and w.p. $(1 - p)$ we sample $(\boldsymbol{x}, y) \sim \mathcal{D}_{\boldsymbol{w}^*}^{\Pi}$. We denote by $f_{\boldsymbol{w}^*}$ to be the target function generating the labels in the distribution $\mathcal{D}_{\boldsymbol{w}^*}$, namely:

$$f_{\boldsymbol{w}^*}(\boldsymbol{x}) = \begin{cases} \sum_{i=1}^{n} w_i^* x_i - \frac{n}{2} & \text{if } x_0 = 1 \\ \prod_{i=1}^{n} x_i^{w_i^*} & \text{if } x_0 = -1 \end{cases}$$

## 2.2 Network Architecture and Training

We consider a DenseNet-like network defined as follows:

$$g_\theta(\boldsymbol{x}) = \boldsymbol{u}^\top \boldsymbol{W} \boldsymbol{x} + \boldsymbol{v}^\top \sigma(\boldsymbol{W} \boldsymbol{x} + \boldsymbol{b})$$

where $\boldsymbol{W} \in \mathbb{R}^{k \times n+1}$, $\boldsymbol{u}, \boldsymbol{v}, \boldsymbol{b} \in \mathbb{R}^k$, and we denote $\theta = (\boldsymbol{W}, \boldsymbol{u}, \boldsymbol{v}, \boldsymbol{b})$.

Denote by $\theta^{(t)}$ the parameters at step $t$ (where $t = 0$ is initialization). We use the square-loss function $\ell(y, \hat{y}) = \frac{1}{2}(y - \hat{y})^2$, and train the network with SGD updates. That is, for some target distribution $\mathcal{D}$ over $\mathcal{X} \times \mathcal{Y}$ and for some choice of step-size sequence $\eta_1, \eta_2, \dots$ we sample $S_t \sim \mathcal{D}^B$ and update,

$$\theta^{(t+1)} = \theta^{(t)} - \eta_t \frac{1}{B} \sum_{(\boldsymbol{x}, y) \in S_t} \nabla_{\theta^{(t)}} \ell\left(g_{\theta^{(t)}}(\boldsymbol{x}), y\right)$$

We allow $\eta_t$-s to be vectors, to facilitate the usage of different step-size and weight-decay for different layers. For compatibility with standard convex-optimization results, we will analyze the loss w.r.t. the averaged parameters $\bar{\theta} = \frac{1}{T} \sum_{t=1}^{T} \theta^{(t)}$.

## 2.3 Results

We start by showing that, when training on the distribution $\mathcal{D}_{\boldsymbol{w}^*}$, i.e., a distribution that contains a mixture of the Task #1 and Task #2, we get a network with low error on the mixture distribution. In other words, training on the $\mathcal{D}_{\boldsymbol{w}^*}$ results in a model that achieves good performance on both tasks.

---

[2]We assume for simplicity that $n$ is even and $|\boldsymbol{w}^*| = n/2$, but the result can be extended to the general case.

**Theorem 1.** *Fix some $\epsilon, \delta \in (0, 1)$, and assume that $n \geq 2\log_2(1/\epsilon)$ There exist an initialization scheme and learning-rate schedule, such that, for every $\boldsymbol{w}^* \in \{0, 1\}^n$, running SGD with batch of size $B = O\left(\frac{n^8}{\epsilon p^2}\right)$ sampled from $\mathcal{D}_{\boldsymbol{w}^*}$, with network size $k \geq n\log(n/\delta)$ for $T = O\left(\frac{n^7}{\epsilon^2}\right)$ iterations returns $\bar{\theta}$ satisfying,*

$$L_{\mathcal{D}_{\boldsymbol{w}^*}}(g_{\bar{\theta}}) \leq \epsilon,$$

*with probability $\geq 1 - 4T\delta$.*

The key to showing the above result lies in the analysis of the gradient of the weights $\boldsymbol{W}$ on the first iteration of SGD. We show that the population gradient already contains the information about the vector $\boldsymbol{w}^*$, indicating which bits are relevant for computing the sum/parity.

**Lemma 2.** *It holds that:*

$$\mathbb{E}_{(\boldsymbol{x}, y) \sim \mathcal{D}_{\boldsymbol{w}^*}}\left[\frac{\partial}{\partial \boldsymbol{W}}\ell(g_{\theta^{(0)}}(\boldsymbol{x}), y)\right] = -p[\boldsymbol{w}, \ldots, \boldsymbol{w}]^\top \in \mathbb{R}^{k \times n+1}$$

*where $\boldsymbol{w} \in \mathbb{R}^{n+1}$ satisfies $w_0 = -\frac{n}{2}$ and for all $i = 1, \ldots, n$, $w_i = w_i^*$.*

To prove Lemma 2, we show that gradient over the Sum distribution $\mathcal{D}_{\boldsymbol{w}^*}^{\Sigma}$ exactly captures $\boldsymbol{w}^*$, while the gradient over the Parity distribution $\mathcal{D}_{\boldsymbol{w}^*}^{\Pi}$ is zero. Therefore, the gradient over the mixture distribution corresponds to $p\boldsymbol{w}^*$ (where the value of $p$ controls what portion of the mixture is given for Task #1). Note that if $p = 0$ (i.e., we only train on the Parity distribution), then the gradient is zero and we cannot learn. Therefore, it is crucial to train on the mixture of the two distributions. The full proof of Lemma 2, as well as Theorem 1, is given in Appendix A

We also note that Theorem 1 shows that learning is possible even for small values of $p$, i.e., when the probability of seeing examples from Task #1 is small. However, there is an extra cost in sample complexity: we need to increase the sample size by a factor of $1/p^2$ to guarantee low error.

Now, if we only observe data from Task #2, learning is provably computationally hard. Indeed, Task #2 is the standard task of learning parities (Blum et al., 2003). This task has been shown to be computationally hard for Statistical Query algorithms (Kearns, 1998) and variants of gradient-descent (Shalev-Shwartz et al., 2017; Abbe & Sandon, 2018). Specifically, learning parities in these different settings has been shown to require the run-time complexity to grow with the size of the target class of parities. In our case, since we consider parities of size exactly $n/2$, we get that the complexity grows with $\binom{n}{n/2} \approx \Theta\left(\frac{2^n}{\sqrt{n}}\right)$. That is, without using any data from Task #1, the complexity of learning grows exponentially with $n$. On the other hand, adding a small number of examples from Task #1 reduces the sample and computational complexity, allowing learning in polynomial time and sample complexity.

## 3 EXPERIMENTS

In this section, we present experimental evidence demonstrating that implicit supervision can facilitate the learning of complex tasks, corroborating our theoretical results. We start by performing experiments on the task of learning parities using a mixture of the Parity/Sum tasks (as defined in Section 2). In line with our theory, we show that standard decoder-only transformers (i.e., GPT-style) can successfully learn parities when they are given access to a mixture distribution containing some examples from the Sum problem, but not when trained only on the Parity problem. Next, we run experiments on the LEGO dataset (Zhang et al., 2023), a synthetic dataset which serves as a proxy for studying logical reasoning tasks. We also experiment with synthetically generated python programs. In both cases, we show that training on a mixture of hard and easy tasks, or training in a multi-label setting where the model outputs labels for easier sub-tasks, significantly improves performance on the harder task.

### 3.1 PARITY MIXTURES

In this subsection, we train a transformer on the Parity/Sum mixture, as introduced in Section 2. More specifically, we will train a GPT-style transformer (see full details in Appendix B) on the distribution $\mathcal{D}_p$ over $\{\pm 1\}^{2n} \times \mathbb{R}$. We generate input and label pairs $(\boldsymbol{x}, y)$ from this distribution by

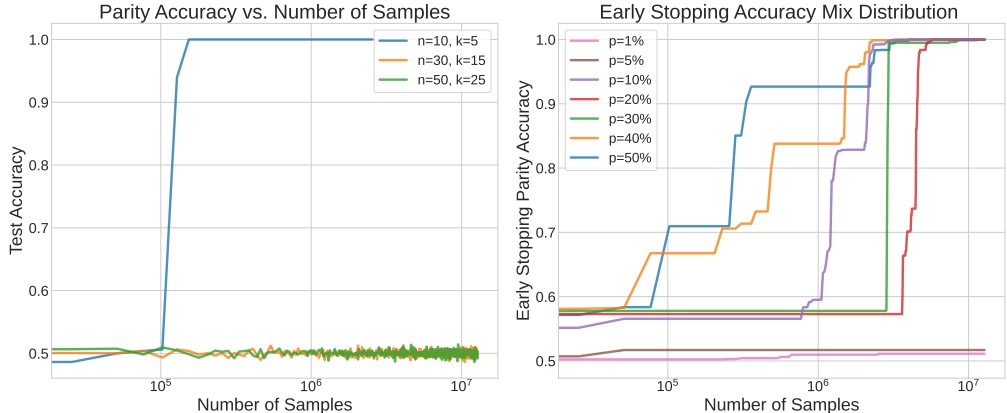

Figure 1: Comparison of model performance on learning parities. (Left) Training with $p = 0$, different values of $n$, and $k = n/2$. The model fails to learn parities as $n$ increases. (Right) Training with a mixture distribution of parities and sums, varying the probability $p$ of selecting a Sum example. The model successfully learns parities when $p$ ranges from 0.1 to 0.5, demonstrating the benefit of implicit supervision.

sampling a Sum example w.p. $p$ and a Parity example w.p. $1 - p$. With a slight deviation from the input definition in Section 2, here $x$ would be a sequence of two $n$-dimensional vectors, where the first is a one-hot vector encoding of the task (i.e., $e_1$ if the task is Sum and $e_2$ if the task is Parity) and the second vector is the raw input sampled from the appropriate distribution ($\mathcal{D}^{\Sigma}_{w^*}$ or $\mathcal{D}^{\Pi}_{w^*}$). Thus, if $x$ was a Parity example then $y = \prod_{i \in w^*} x_i$ and otherwise $y = \frac{1}{n} \sum_{i \in w^*} x_i$ for a Sum example.

A-priori, we are only complicating matters by adding another task for the network to solve. And yet, since the subset corresponding to $w^*$ is shared between the tasks, the network can learn to reuse the "same features" for both tasks. In Figure 1 (Left) we train the model only the Parity distribution $\mathcal{D}^{\Pi}_{w^*}$ and vary the size of $n$ (and recall that the subset size is $k = n/2$). As expected, quite quickly (e.g., for $n = 30$, $k = 15$) the model fails completely, achieving chance accuracy of $50\%$.

**Results.** In Figure 1 (Right), we train the model (with $n = 50$ and $k = 25$) and observe that when using a mixture distribution with $p$ ranging from $0.5$ down to $0.1$, the model is able to learn the parity to full accuracy. So even with a mixture containing only $10\%$ of data from the Sum distribution allows us to crack the hard task of learning parities. We hypothesise that learning parities is also possible with smaller fractions, but requires more compute and data.

### 3.2 LEGO

In this subsection, we demonstrate that adding implicit multi-label supervision is greatly beneficial for obtaining strong reasoning capabilities. Specifically, we train a BERT-style transformer on two variants of the systemic reasoning dataset LEGO (Zhang et al., 2023). The task in this dataset involves performing a series of 12-variable assignments that may also include negation operations. An input example looks like this:

$$j = -\mathtt{f}; \mathtt{f} = -\mathtt{b}; \mathtt{y} = +\mathtt{t}; \mathtt{o} = +\mathtt{e}; \mathtt{d} = +\mathtt{y}; \mathtt{v} = +\mathtt{d};$$

$$\mathtt{h} = -\mathtt{o}; \mathtt{b} = -\mathtt{i}; \mathtt{i} = +1; \mathtt{t} = +\mathtt{l}; \mathtt{e} = -\mathtt{j}; \mathtt{l} = -\mathtt{h};$$

We train on two variants of this problem, corresponding to a single-label and multi-label training. In the single-label variant, the model aims to predict the value of the final variable in the logical chain of assignments. The value of this variable can only be deduced after the values of all the other variables have been successfully inferred (e.g., $\mathtt{v}$ in the example above). In the multi-label task, the model needs to output the assignment of all variables simultaneously. Note that some variables are immediately inferable (they are just assigned a binary value), while others require resolving a long sequence of dependencies. Thus, training in the multi-label setting allows the model to use it's prediction for easily derivable variables to solve the assignment for variables that are hard to infer. See Zhang et al. (2023) for further details on how the LEGO dataset was generated.

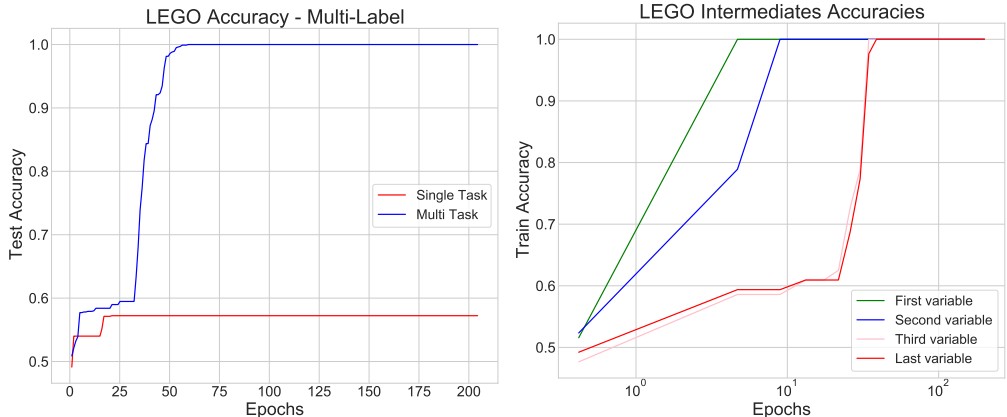

Figure 2: Comparison of model early stopping performance on the synthetic reasoning task LEGO. (Left) Training the model in a single-label setting versus training it in a multi-label setting. The model learns the LEGO task almost perfectly in the multi-label setting, while its performance is close to random-guess in the single-label setting. (Right) Multi-label training leads to implicit curriculum learning in which the model learns to perform the series of assignments one at a time. For clarity, we have omitted the fourth through eleventh curves, as they all converge at a similar rate as the latest (twelfth) variable.

Figure 2 shows that in the variant corresponding to the single-label training, the model is unable to solve the task and reaches only random-guess performance. Meanwhile, the variant corresponding to the multi-label task exhibits an implicit curriculum learning which enables it to achieve almost perfect accuracy.

## 3.3 SYNTHETIC CODE INTERPRETATION

Transitioning to our final task, we address the challenging problem of code interpretation. Nye et al. (2021) demonstrated that adding a scratchpad to a unidirectional GPT-style transformer is a key component for performing this type of task. In this section, we present an alternative architecture consisting of a bidirectional BERT-style transformer that is able to perform code interpretation when trained with either a multi-task mixture or multi-label data. Notably, unlike the scratchpad which requires labeling intermediate results for the whole training dataset, the multi-task mixture requires intermediate labels only for a small portion of the examples.

For this task, we train our BERT-style transformer on different variants of a code interpretation task. This task entails executing a series of Python-like code operations, where each operations is either assignment, addition, subtraction, multiplication or division. For simplicity, all operations are performed in a modulo ten numerical system. For example, here is a possible input code:

$$x = 7; x \mathrel{-}= 1; x \mathrel{*}= 3; x \mathrel{//}= 2; x \mathrel{+}= 5;$$

We train on three variants of this problem: 1) Single Task; 2) Multi-Label; and 3) Mixture of Single Task with intermediate supervision. In the single-task/single-label setting, the model must determine the final value of the variable x after executing all the lines of code. In the second setup, which corresponds to a multi-label setting, the model must output *all* the intermediate values of the variable, that is, output the value of x after each line of code is executed. In the last setting, we combine the single-task and multi-label distribution by augmenting the single task with intermediate supervision of each task with small probability (e.g., $\frac{1}{300}$). Thus, the first step is supervised once every 300 examples in expectation as is the second step, and so on. The last output's label, however, is seen in every sample. Due to the varying number of labels between samples of the third variant, we added a [CLS] token for inputs in this task, while employing a token classification approach for the multi-label variant. To ensure a fair comparison, we compared each of these variants to a single task with its corresponding classification approach.

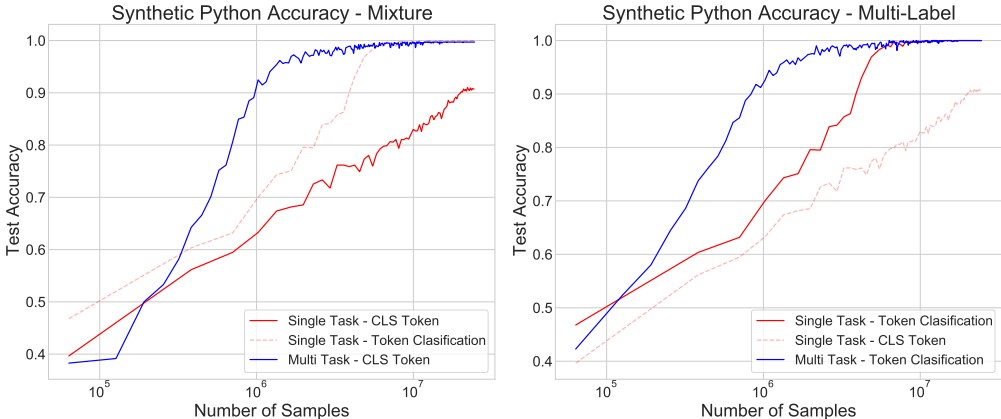

Figure 3: Comparison of model performance on synthetic python. (Left) Training with $p = 0$ and $p \approx 3\%$. The model learns synthetic python much faster when $p > 0$ even for very small $p$. (Right) Training with a multi-label multi task comparing to a single task based training. The model converge faster with the multi-label setup, demonstrating the benefit of implicit supervision.

Figure 3 shows that the model converges much faster in the multi-task and multi-label setting, compared to the single-task/single-label setting where the model only predicts the final output. We note that unlike previous cases, here a model trained only on the final is able to converge to high accuracy after enough training. However, training in the multi-task or multi-label setting clearly improves convergence time and sample complexity.

## 4    DISCUSSION

LLMs are demonstrating increasingly remarkable capabilities in a wide range of tasks, including mathematical problem-solving, code generation, and writing assistance. The current state-of-the-art approach to achieving high performance in LLMs involves a two-step process: pretraining on a vast and diverse corpus of data (often referred to as "the entire internet") and finetuning on a specific downstream task using a combination of supervised learning, reinforcement learning from human feedback (RLHF), and instruction finetuning. Despite the unparalleled performance of this pipeline, the pretraining phase entails substantial computational costs, raising questions about the necessity of such extensive pretraining for models designed to perform tasks within a specific domain. For instance, if the primary objective is to solve mathematical problems, is it essential to train a model on a large corpus of books, Wikipedia articles, and arbitrary code from GitHub?

Our research offers a partial response to these questions by demonstrating that training on a mixture of multiple tasks can provide the model with implicit intermediate supervision, enabling it to perform better on the downstream task. Our theoretical findings reveal instances where training exclusively on Task A is unsuccessful, but "pretraining" on a combination of Task A and Task B yields favorable results *on both tasks*. In such cases, bypassing the pretraining phase and focusing solely on the final task would inevitably lead to failure. Our empirical results further support this notion.

We posit that our findings capture, to a certain extent, the underlying principles that render pretraining an indispensable and inescapable step in the development of LLMs. We hope that these insights will contribute to a deeper understanding of how the training schemes of language models influence their downstream performance and inform future research on more efficient and effective training methodologies.

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

## A  PROOFS

We assume the following initialization: $\boldsymbol{u}^{(0)} = [1, \ldots, 1]$, $\boldsymbol{W}^{(0)} = 0$, $\boldsymbol{v}^{(0)} = 0$ and $b_i^{(0)} \sim \text{Uniform}(\{\beta_0, \ldots, \beta_{n/2}\})$ where $\beta_i = p \cdot (-2n/2 + 2i + 1)$. At the first step, we use step-size of $\eta_0 = 1$ for all layers.

Throughout the proofs, we use the notation $\boldsymbol{u}^{(t)}$ to indicate the value of the parameter $\boldsymbol{u}$ at iteration $t$, but omit this index for iteration $t = 0$ (i.e., $\boldsymbol{u} = \boldsymbol{u}^{(0)}$, and similarly for other parameters).

**Lemma 3.** *For every subset of examples $S$ we have:*

$$\mathbb{E}_{(\boldsymbol{x},y) \sim S} \left[ \frac{\partial}{\partial \boldsymbol{u}, \boldsymbol{v}, \boldsymbol{b}} \ell(g_{\theta^{(0)}}(\boldsymbol{x}), y) \right] = 0$$

*Proof.* Observe that $g_\theta(\boldsymbol{x}) = 0$ for all $\boldsymbol{x}$, therefore:

$$\nabla_\theta \ell(g_\theta(\boldsymbol{x}), y) = (g_\theta(\boldsymbol{x}) - y) \nabla_\theta g_\theta(\boldsymbol{x}) = -y \nabla_\theta g_\theta(\boldsymbol{x})$$

From the initialization of $\boldsymbol{v} = 0$ and $\boldsymbol{W} = 0$ we get:

$$\frac{\partial}{\partial \boldsymbol{u}, \boldsymbol{v}, \boldsymbol{b}} g_\theta(\boldsymbol{x}) = 0$$

and therefore, for every $S$,

$$\mathbb{E}_{(\boldsymbol{x},y) \sim S} \left[ \frac{\partial}{\partial \boldsymbol{u}, \boldsymbol{v}, \boldsymbol{b}} \ell(g_{\theta^{(0)}}(\boldsymbol{x}), y) \right] = 0$$

$\square$

*Proof of Lemma 2.* Since at initialization $\boldsymbol{v} = 0$ and $\boldsymbol{u} = 1$ we get:

$$\frac{\partial}{\partial \boldsymbol{W}} g_\theta(\boldsymbol{x}) = \boldsymbol{u} \boldsymbol{x}^\top + \boldsymbol{v} \sigma'(\boldsymbol{W}\boldsymbol{x} + \boldsymbol{b}) \boldsymbol{x}^\top = \begin{bmatrix} \boldsymbol{x}^\top \\ \vdots \\ \boldsymbol{x}^\top \end{bmatrix}$$

Therefore, we get, for all $j \in [k]$

$$\mathbb{E}_{(\boldsymbol{x},y) \sim \mathcal{D}_{\boldsymbol{w}^*}} \left[ \frac{\partial}{\partial \boldsymbol{W}_j} \ell(g_{\theta^{(0)}}(\boldsymbol{x}), y) \right] = -\mathbb{E}_{(\boldsymbol{x},y) \sim \mathcal{D}_{\boldsymbol{w}^*}} [y\boldsymbol{x}]$$

We first analyze the last term for the sum distribution $\mathcal{D}_{\boldsymbol{w}^*}^\Sigma$:

$$\mathbb{E}_{\mathcal{D}_{\boldsymbol{w}^*}^\Sigma} [y\boldsymbol{x}_i] = \mathbb{E}_{\mathcal{D}_{\boldsymbol{w}^*}^\Sigma} \left[ x_i \cdot \sum_{i=1}^n w_l^* x_l - x_i \cdot \frac{n}{2} \right] = \begin{cases} -\frac{n}{2} & \text{if } i = 0 \\ w_i^* & \text{if } i > 0 \end{cases}$$

We analyze the same term for the Parity distribution $\mathcal{D}_{\boldsymbol{w}^*}^{\Pi}$:

$$\mathbb{E}_{\mathcal{D}_{\boldsymbol{w}^*}^{\Pi}}[y\boldsymbol{x}_i] = \mathbb{E}_{\mathcal{D}_{\boldsymbol{w}^*}^{\Pi}}\left[x_i \cdot \prod_l x_l^{w_i^*}\right] = 0$$

and the required follows from the above. $\qquad\square$

**Lemma 4.** *Fix some $n'$. For every $\boldsymbol{x} \in \{\pm 1\}^{n'}$ denote $s(\boldsymbol{x}) = \sum_{i=1}^{n'} x_i$ and $\pi(\boldsymbol{x}) = \prod_{i=1}^{n'} x_i$. Let $\beta_0', \dots, \beta_{n'}'$ s.t. $\beta_i' = -2n' + 2i + 1$. There exists a choice of $\nu_0, \dots, \nu_{n'}$, with $|\nu_{n'}| \leq 4n'$, such that:*

$$\sum_{i=0}^{n'} \nu_i \sigma(s(\boldsymbol{x}) + \beta_i' + n') = \pi(\boldsymbol{x}) - s(\boldsymbol{x}) - n'$$

*Proof.* If $n'$ is even, we choose $v_{n'} = 1$ and $v_i = (-1)^i \cdot (4(n' - i) + 2)$ for $i < n'$. If $n'$ is odd, we choose $v_{n'} = -1$ and $v_i = (-1)^i \cdot (4(n' - i) - 2)$ for $i < n'$. $\qquad\square$

**Lemma 5.** *Define $\beta_0, \dots, \beta_{n/2}$ s.t. $\beta_i = p \cdot \beta_i'$ (where $\beta_i'$ is as defined in Lemma 4). Let $\boldsymbol{w}$ as defined in Lemma 2. Define $\psi_i(\boldsymbol{x}) = \sigma(\langle p\boldsymbol{w}, \boldsymbol{x}\rangle + \beta_i)$. Then, there exists $v_0', \dots, v_{n/2+1}' \in \mathbb{R}$ with $|v_i| \leq 2n/p$ s.t. for all $\boldsymbol{x} \in \mathcal{X}$ s.t. $\boldsymbol{x} \neq \boldsymbol{1}$ it holds that*

$$\tilde{f}(\boldsymbol{x}) := \sum_{i=0}^{n/2} v_i' \psi_i(\boldsymbol{x}) + v_{n/2+1}' \langle p\boldsymbol{w}, \boldsymbol{x}\rangle = f_{\boldsymbol{w}^*}(\boldsymbol{x})$$

*Proof.* Let $n' = n/2$ and let $\nu_0, \dots, \nu_{n/2+1}$ the parameters given from Lemma 4. For every $\boldsymbol{x} \in \mathcal{X}$, denote $\boldsymbol{x}_{\boldsymbol{w}^*} \in \{\pm 1\}^{n/2}$ the value of the bits of $\boldsymbol{x}$ in the active coordinates of $\boldsymbol{w}^*$. Denote $v_i' = \frac{1}{p}\nu_i$ for every $0 \leq i \leq n$, and let $v_{n/2+1}' = \frac{1}{p}$. Observe that by definition of $\boldsymbol{w}$ we have:

$$\sum_{i=0}^{n/2} v_i' \psi_i(\boldsymbol{x}) + v_{n+1}' \langle p\boldsymbol{w}, \boldsymbol{x}\rangle = \frac{1}{p} \sum_{i=0}^{n/2} \nu_i \sigma(p\langle \boldsymbol{w}, \boldsymbol{x}\rangle + \beta_i) + pv_{n/2+1} \langle \boldsymbol{w}, \boldsymbol{x}\rangle$$

$$= \frac{1}{p} \sum_{i=0}^{n/2} \nu_i \sigma\left(ps(\boldsymbol{x}_{\boldsymbol{w}^*}) - \frac{pn}{2}x_0 + p\beta_i'\right) + s(\boldsymbol{x}_{\boldsymbol{w}^*}) - \frac{n}{2}x_0$$

$$= \sum_{i=0}^{n/2} \nu_i \sigma\left(s(\boldsymbol{x}_{\boldsymbol{w}^*}) - \frac{n}{2}x_0 + \beta_i'\right) + s(\boldsymbol{x}_{\boldsymbol{w}^*}) - \frac{n}{2}x_0$$

We observe two cases:

- Assume $x_0 = 1$. Then, for all $\boldsymbol{x} \neq \boldsymbol{1}$ we have $s(\boldsymbol{x}_{\boldsymbol{w}^*}) - n/2 + \beta_i' < 0$, and therefore:

$$\tilde{f}(\boldsymbol{x}) = s(\boldsymbol{x}_{\boldsymbol{w}^*}) - \frac{n}{2} = f_{\boldsymbol{w}^*}(\boldsymbol{x})$$

- Assume $x_0 = -1$. Then, using Lemma 4 we have:

$$\tilde{f}(\boldsymbol{x}) = \sum_{i=0}^{n/2} \nu_i \sigma\left(s(\boldsymbol{x}_{\boldsymbol{w}^*}) + \frac{n}{2} + \beta_i'\right) + s(\boldsymbol{x}_{\boldsymbol{w}^*}) + \frac{n}{2} = \pi(\boldsymbol{x}_{\boldsymbol{w}^*}) = f_{\boldsymbol{w}^*}(\boldsymbol{x})$$

$\qquad\square$

**Lemma 6.** *Assume that $\boldsymbol{b}^{(1)} = \boldsymbol{b}^{(0)}$ and*

$$\left\| \boldsymbol{W}^{(1)} - \mathbb{E}_{(\boldsymbol{x},y)\sim\mathcal{D}_{w^*}} \left[ \frac{\partial}{\partial \boldsymbol{W}} \ell(g_{\theta^{(0)}}(\boldsymbol{x}), y) \right] \right\|_\infty \leq \tau$$

*Fix some $\delta \in (0,1)$. Then, if $k \geq n \log(n/\delta)$, with probability at least $1 - \delta$, there exists $\boldsymbol{u}, \boldsymbol{v} \in \mathbb{R}^k$ s.t. $\|\boldsymbol{u}\|_2^2, \|\boldsymbol{v}\|_2^2 \leq 4n^3/p^2$ and*

$$g(\boldsymbol{x}) = \boldsymbol{u}^\top \boldsymbol{W}^{(1)} \boldsymbol{x} + \boldsymbol{v}^\top \sigma(\boldsymbol{W}^{(1)} \boldsymbol{x} + \boldsymbol{b}^{(1)})$$

*satisfies $L_{\mathcal{D}_{w^*}} [\ell(g(\boldsymbol{x}), y)] \leq \frac{8n^6\tau^2}{p^2} + 2^{-n/2}$.*

*Proof.* Denote by $\boldsymbol{W}_j^{(1)}$ the $j$-th column of $\boldsymbol{W}$ after the first update. From Lemma 2, together with the choice of initialization, we get that, for $\boldsymbol{w}$ as defined in Lemma 2, for all $j$,

$$\left\| \boldsymbol{W}_j^{(1)} - p\boldsymbol{w} \right\|_\infty \leq \tau$$

For some $i \in \{0, \ldots, n/2\}$, the probability that there exists $j \in [k]$ s.t. $b_j = \beta_i$ is:

$$1 - \Pr\left[\forall_j b_j \neq \beta_i\right] = 1 - (1 - \frac{1}{n/2 + 1})^k \geq 1 - e^{-\frac{k}{(n/2+1)}} \geq 1 - \frac{\delta}{n}$$

And from the union bound, w.p. at least $1 - \delta$, for all $i \in \{0, \ldots, n/2\}$ there exists $j \in [k]$ s.t. $b_j = \beta_i$. For simplicity of notation, and w.l.o.g., we assume that $b_i = \beta_i$ for all $i \in \{0, \ldots, n/2\}$.

Now, let $v_0', \ldots, v_{n/2+1}'$ as defined in Lemma 5, and let $\boldsymbol{v}, \boldsymbol{u}$ s.t. $v_i = v_i'$ for all $i \in [n/2]$ and $v_i = 0$ for all $i > n/2$. Additionally, and let $u_i = \frac{1}{k} v_{n/2+1}$ for all $i$. Denote $\tilde{\psi}_i(\boldsymbol{x}) = \sigma(\boldsymbol{W}_i^{(1)} \boldsymbol{x} + b_i)$ and observe that, for all $\boldsymbol{x}$,

$$\left| \psi(\boldsymbol{x}) - \widetilde{\psi}(\boldsymbol{x}) \right| = \left| \sigma(\langle p\boldsymbol{w}, \boldsymbol{x} \rangle + \beta_i) - \sigma\left( \left\langle p\boldsymbol{W}_i^{(1)}, \boldsymbol{x} \right\rangle + \beta_i \right) \right|$$
$$\leq \left| \left\langle p\boldsymbol{w} - \boldsymbol{W}_i^{(1)}, \boldsymbol{x} \right\rangle \right| \leq n\tau$$

Observe that, for $\tilde{f}$ as defined in Lemma 5

$$\left| g(\boldsymbol{x}) - \tilde{f}(\boldsymbol{x}) \right| \leq \sum_{i=0}^{n/2} |v_i'| \left| \widetilde{\psi}_i(\boldsymbol{x}) - \psi_i(\boldsymbol{x}) \right| + |v_{n/2+1}'| \left| \left\langle \frac{1}{k} \sum_{j\in[k]} \boldsymbol{W}_j^{(1)} - p\boldsymbol{w}, \boldsymbol{x} \right\rangle \right|$$
$$\leq \frac{2n^3\tau}{p} + \frac{2n^2\tau}{p} \leq \frac{4n^3\tau}{p}$$

From Lemma 5, for all $\boldsymbol{x} \neq \boldsymbol{1}$ we have $\tilde{f}(\boldsymbol{x}) = f_{\boldsymbol{w}^*}(\boldsymbol{x})$. Since $\tilde{f}(\boldsymbol{x}) < 2^{n/4}$ we get

$$\mathbb{E}_{\mathcal{D}_{w^*}} [\ell(g(\boldsymbol{x}), y)] = \frac{1}{2} \mathbb{E}_{\mathcal{D}_{w^*}} \left[ (g(\boldsymbol{x}) - f_{\boldsymbol{w}^*}(\boldsymbol{x}))^2 \right] \leq \frac{1}{2} \left( 4n^3\tau/p \right)^2 + 2^{-n/2} x$$

$\square$

We use the following Theorem (Theorem 14.13 from Shalev-Shwartz & Ben-David (2014)).

**Theorem 7.** *Assume the loss function $\ell(\cdot, z)$ is convex, $\alpha$-smooth, and nonnegative. Then, if we run the SGD algorithm with step-size $\eta$ for minimizing $L_{\mathcal{D}}(\boldsymbol{u})$ we have that for every $\boldsymbol{u}^\star$,*

$$L_{\mathcal{D}}(\bar{\boldsymbol{u}}) \leq \frac{1}{1 - \eta\alpha} \left( L_{\mathcal{D}}(\boldsymbol{u}^\star) + \frac{\|\boldsymbol{u}^\star\|^2}{2\eta T} \right)$$

*with probability of at least $1 - T \cdot 2\delta$ where $\bar{\boldsymbol{u}} = \frac{1}{T} \sum_{t=1}^T \boldsymbol{u}^{(t)}$*

*Proof of Theorem 1.* We show the following claims,

**Claim 1**: If $B \geq \frac{n^2}{2\tau^2} \log(2nk/\delta)$ then w.p. at least $1 - \delta$ we have:

$$\left\| \boldsymbol{W}^{(1)} - \mathbb{E}_{(\boldsymbol{x},y) \sim \mathcal{D}_{w^*}} \left[ \frac{\partial}{\partial \boldsymbol{W}} \ell(g_{\theta^{(0)}}(\boldsymbol{x}), y) \right] \right\|_\infty \leq \tau$$

**Proof**: Note that for every $i, j$ and $(\boldsymbol{x}, y) \sim \mathcal{D}$ we have $\left| \frac{\partial}{\partial \boldsymbol{W}_{i,j}} \ell(g_\theta(\boldsymbol{x}), y) \right| = |yx_j| \leq n$. Now, observe that $\boldsymbol{W}^{(1)} = \frac{1}{B} \sum_{(\boldsymbol{x},y) \in S} \frac{\partial}{\partial \boldsymbol{W}} \ell(g_\theta(\boldsymbol{x}), y)$. Therefore, using Hoeffding's inequality

$$P\left( \left| \boldsymbol{W}_{i,j}^{(1)} - \mathbb{E}[\boldsymbol{W}_{i,j}^{(1)}] \right| \geq \tau \right) \leq 2 \exp\left( \frac{-2B\tau^2}{n^2} \right) \leq \frac{\delta}{nk}$$

and the required follows from the union bound.

Let $\Psi, \Phi : \mathcal{X} \to \mathbb{R}^k$ s.t. $\Psi(\boldsymbol{x}) = \boldsymbol{W}^{(1)}\boldsymbol{x}$ and $\Phi(\boldsymbol{x}) = \sigma(\boldsymbol{W}^{(1)}\boldsymbol{x} + \boldsymbol{b}^{(1)})$. For any $\boldsymbol{u}, \boldsymbol{v}$ denote
$$L_{\mathcal{D}_{w^*}}(\boldsymbol{u}, \boldsymbol{v}) = \mathbb{E}_{\mathcal{D}_{w^*}} \left[ \ell(\langle (\boldsymbol{u}, \boldsymbol{v}), (\Psi(\boldsymbol{x}), \Phi(\boldsymbol{x})) \rangle, y) \right]$$

**Claim 2**: With probability at least $1 - 2\delta$, if we choose $B \geq \frac{n^2}{2\tau^2} \log(2nk/\delta)$ and $k \geq n \log(n/\delta)$ then for all $\boldsymbol{x}$, $\|(\Psi(\boldsymbol{x}), \Phi(\boldsymbol{x}))\|_2 \leq (4p + 2\tau)n^2$ and also there exist $\boldsymbol{u}^\star, \boldsymbol{v}^\star$ with $\|(\boldsymbol{u}^\star, \boldsymbol{v}^\star)\|_2^2 \leq 8n^3/p^2$ s.t. $L_{\mathcal{D}_{w^*}}(\boldsymbol{u}^\star, \boldsymbol{v}^\star) \leq \frac{8n^6\tau^2}{p^2} + 2^{-n/2}$.

**Proof**: First, from Claim 1 we have, for all $i$
$$|\Psi_i(\boldsymbol{x})| = \left| \left\langle \boldsymbol{W}_i^{(1)}, \boldsymbol{x} \right\rangle \right| \leq p |\langle \boldsymbol{w}, \boldsymbol{x} \rangle| + \left\| \boldsymbol{w} - \boldsymbol{W}_i^{(1)} \right\|_\infty \|\boldsymbol{x}\|_1 \leq (p + \tau)n$$
additionally, for all $i$
$$|\Phi_i(\boldsymbol{x})| = \left| \sigma\left( \left\langle \boldsymbol{W}_i^{(1)}, \boldsymbol{x} \right\rangle + \boldsymbol{b}^{(1)} \right) \right| \leq \left| \left\langle \boldsymbol{W}_i^{(1)}, \boldsymbol{x} \right\rangle \right| + \left| \boldsymbol{b}^{(1)} \right| \leq (p + \tau)n + 2pn$$

and the required follows using Lemma 6, using the union bound.

**Proof of the Theorem**: Let $\tau^2 = \frac{p^2\epsilon}{8n^6}$ and assume $n \geq 2\log_2(1/\epsilon)$. From the above claim if $B \geq \frac{4n^8}{p^2\epsilon}$ and $k \geq n \log(n/\delta)$, w.p. at least $1 - 2\delta$ it holds for all $\boldsymbol{x}$ that $\ell(\langle (\boldsymbol{u}, \boldsymbol{v}), (\Psi(\boldsymbol{x}), \Phi(\boldsymbol{x})) \rangle, y)$ is $\alpha$-smooth with $\alpha \leq (4p + 2\tau)^2 n^4 \leq 36p^2n^4$ and there exist $\boldsymbol{u}^\star, \boldsymbol{v}^\star$ with $\|(\boldsymbol{u}^\star, \boldsymbol{v}^\star)\|_2^2 \leq 8n^3/p^2$ s.t. $L_{\mathcal{D}_{w^*}}(\boldsymbol{u}^\star, \boldsymbol{v}^\star) \leq 2\epsilon$. Assume that after the first update step we use $\eta = \frac{1}{36p^2n^4(1+3/\epsilon)}$ and run for $T \geq \frac{12 \cdot 8 \cdot 36n^7}{\epsilon^2}$ steps, updating only the second layer. Then, from Theorem 7 it holds that
$$L_{\mathcal{D}_{w^*}}(\bar{\boldsymbol{u}}, \bar{\boldsymbol{v}}) \leq 3\epsilon$$
with probability of at least $1 - T \cdot 2\delta$. $\qquad\square$

## B EXPERIMENTAL DETAILS

### B.1 PARITY MIXTURES

We use GPT-2-small models (Radford et al., 2018) as implemented by HuggingFace with embedding dimension 128, 6 layers and 4 attention heads (total 1.2M paramteres). We train them using Adam with learning rate of 0.0002 and batch size of 256, training for up to 100000 train steps. We use Mean Squared Error loss for the Sum task and Binary Cross Entropy for the Parity task.

### B.2 LEGO

We use BERT-Base models (Devlin et al., 2019) as implemented by HuggingFace with embedding dimension 768, 12 layers, 12 attention heads and the standard BERT vocabulary (total 108M paramteres). We train them using Adam with a batch size of 1024, wight decay 0.01 and the best learning rate among of $10^{-5}$, $2.5 \cdot 10^{-5}$, $5 \cdot 10^{-5}$, $10^{-4}$, $2.5 \cdot 10^{-4}$, $5 \cdot 10^{-4}$. We used a cosine annealing learning rate scheduler with 1000 warmup steps and trained for 200 epochs on a generated dataset of $120,000$ synthetic samples each one composed of 12 variable assignments. We use the Cross Entropy loss for both the single task and the multi-label variants of this task.

## B.3 SYNTHETIC CODE INTERPRETATION

We use BERT-Base models (Devlin et al., 2019) as implemented by HuggingFace with embedding dimension 768, 12 layers, 12 attention heads and a small vocabulary size of 16 tokens (total 85M paramteres). We train them using Adam with a batch size of 1024, no wight decay and the best learning rate among of $10^{-5}$, $2.5 \cdot 10^{-5}$, $5 \cdot 10^{-5}$, $10^{-4}$, $2.5 \cdot 10^{-4}$, $5 \cdot 10^{-4}$. We used a cosine annealing learning rate scheduler without warmup steps and trained for $24,000$ iteration that corresponds to a single epoch on a synthetic samples each one composed of 10 code statements. In the mixture of multi-task we augmented the single task with intermediate supervision of each task with a probability of $1/320$. We use the Cross Entropy loss for all the variants of this task.

