# OpenReview forum: "Implicit Intermediate Supervision for Learning Complex Functions"
_ICLR.cc/2024/Conference — Submitted to ICLR 2024_

### Official Review · Reviewer_vLEV · 2023-10-30

**Soundness:** 2 fair
**Presentation:** 3 good
**Contribution:** 3 good
**Rating:** 5
**Confidence:** 3

**Summary:**

In this work, the authors explore the effect of explicit intermediate supervision on learning complex functions. Specifically, they find that training on a mixture of multiple tasks yields better results than training only on one task. Some theoretical and empirical results show the effects of a combination of two tasks. Their findings could imply better training schemes for language models.

**Strengths:**

1. This is an interesting work on the investigation of learning effects with a mix of tasks. Some theoretical and empirical evidence is shown for the learning effect. The results on a mixture of the Parity/Sum task are interesting

2. They also empirically show that it is easier and faster for the learner if the signals from easily inferred labels to learn target are provided. Experiments on LEGO and code interpretation task are done.

3. Their findings on learning complex tasks contribute to the understanding of large language model learning and provide valuable insights for future related work on efficient training.

**Weaknesses:**

1. The theoretical results on the Parity/Sum task reply to some strong assumptions: bilinear parameterization, some initialization (for example, v = 0). Under these assumptions, the gradient over the parity distribution samples is zero. The assumption looks a little strong.

2. It would be to show more experiment results for some settings, for example, performance on Sum task when training with a mixture distribution, or more Sum task samples and fewer Parity task samples ( p range from 0.5 to 1. In Figure 1, results with p ranges from 0.1 to 0.5 are shown.

**Questions:**

Typo errors:

In section 2.2, the parameter W size should be R*{k x (n + 1)}, not R*{k x n + 1}.

---

> ### Author Response · Authors · 2023-11-20
>
> We would like to thank the reviewer for their time reviewing our work. We address the weaknesses below.
> 1. Indeed, our architecture and initialization are somewhat non-standard. We use this setting in order to simplify our theoretical analysis, but we believe that similar results also hold for more “standard” networks, at the cost of making the analysis much more involved. Specifically, we would like to note that while our initialization ensures that the gradient over the parity distribution (at initialization) is exactly zero, it can be easily shown (e.g., using an analysis similar to [1]) that for any choice of architecture and initialization, the gradient over the parity is extremely small (that is, exponentially close to zero). Also note that we also report empirical experiments with “standard” choices of network architectures, which validate our theoretical findings.
> 2. Thank you for your suggestions. These experiments can indeed contribute to the paper, and we will add them to the final version of the manuscript.
>
> Please consider raising your score if you feel that your concerns were answered, and if not we are happy to continue discussing your comments.
>
> [1] https://proceedings.mlr.press/v70/shalev-shwartz17a/shalev-shwartz17a.pdf

---

### Official Review · Reviewer_rg2R · 2023-10-30

**Soundness:** 2 fair
**Presentation:** 2 fair
**Contribution:** 2 fair
**Rating:** 5
**Confidence:** 4

**Summary:**

This paper proposes a new supervision named implicit intermediate supervision for complex functions. Particularly the authors provide theoretical and empirical evidence to support the effectiveness of implicit supervision.

**Strengths:**

1. It is novel that the paper pays attention implicit intermediate supervision instead of explicit supervision to solve intricate tasks in language modeling, and it provides detailed proof of the notion.

2. The paper could also contribute to the understanding of how large language models learn complex tasks and may facilitate research on more efficient and effective training methodologies.

**Weaknesses:**

1. There is something wrong with the structure of this paper, for example, section 1.2 is named as Related Work, which is usually a separate chapter. Moreover, this paper losses the Conclusion part.

2. This paper does not contain an example to show the advantages of implicit supervision over explicit supervision, or an example that demonstrates how the implicit supervision work. So can the authors provide a figure in Section 1 that can make readers get your innovation quickly?

3. The authors point out that explicit intermediate step-by-step supervision is time consuming compared to the implicit supervision in abstract, but they do not provide experimental results to verify this view.

**Questions:**

Assuming that there are now n tasks, task 1, task 2, …, task n. Will the following scenario occur: training task 1 with data from task 1 to task i yields excellent results, but when training task 1 with data from task 1 to task i+1, the performance significantly deteriorates?

---

> ### Author Response · Authors · 2023-11-20
>
> Thank you for your valuable feedback and comments.
> Weaknesses:
>
> We will restructure the paper to have a separate Related Works section, and extend the discussion section to include conclusions. We do note that it is very common in our field to have the Related Works as a subsection of the introduction, and to have a combined Discussion and Conclusions section.
>
> The main claim in the paper is that intermediate supervision, which is required for solving some hard tasks, can happen naturally by training on a mixture of tasks. For example, computing the parity function over some subset of bits is hard to learn by itself, but when we train on a mixture of the parity and the sum task (where the goal is to compute the sum of the bits in the subset), learning becomes easy. We call this type of supervision implicit, because nowhere in the training process did we “force” the network to use its ability to compute the sum in order to compute the parity. In contrast, to provide explicit supervision, we could for example construct a dataset which includes “chain-of-thought” reasoning that forces the model to first compute the sum, and then use this result to compute the parity. We claim that implicit supervision appears more naturally in data “in-the-wild” (for example, data randomly scraped from the internet). Such data is very likely to contain many different tasks which can positively contribute to each other, but might not include clean step-by-step supervision for particular problems of interest. Generating such step-by-step supervision usually requires costly human annotations. In this sense, we claim that implicit supervision is “better”. We will update the manuscript to clarify this distinction between implicit and explicit intermediate supervision.
>
> Regarding your question: note that we do not argue that all tasks help each other, and it is indeed possible that training on one task harms the performance of another task. Our focus is mainly on showing that in some cases, one task can dramatically improve the performance of another task.
>
> Please consider raising your score if you feel that your concerns were answered, and if not we are happy to continue discussing your comments.

---

### Official Review · Reviewer_JC1K · 2023-10-31

**Soundness:** 3 good
**Presentation:** 3 good
**Contribution:** 3 good
**Rating:** 5
**Confidence:** 3

**Summary:**

The paper focuses on intermediate supervision. Popular approaches mainly use explicit step-by-step supervision, whereas the authors investigate implicit step-by-step supervision. The motivation is that models can implicitly explore the structure of tasks, and easy tasks can benefit the understanding of hard tasks. The paper then proposes two settings, multi-tasks and multi-labels, and uses a synthetic parity learning problem and feed-forward network structure to theoretically justify the motivation. During experiments, authors conduct experiments on transformer architectures and other datasets to show the generality of the proposed approach.

**Strengths:**

1. The paper explains implicit intermediate supervision, which may help understand the large language model's capability of solving complex problems.
2. The paper provides both theoretical understanding and empirical justification. The experiments show the observation also applies to the Transformer architecture.

**Weaknesses:**

1. The theory is a bit limited to the specific synthetic task. Are there possibilities to extend the idea to support a more general case?
2. The experiments are based on synthetic datasets without realistic datasets. I think datasets that can be used in curriculum learning can be used here too, to justify the applicability of the proposed approach.

I would currently put my score to 5 but may change my score based on the authors' rebuttal.

**Questions:**

Please see the weakness part.

---

> ### Author Response · Authors · 2023-11-20
>
> We would like to thank the reviewer for their time reviewing our work. We address the weaknesses below.
>
> 1. Since our theoretical understanding of deep learning is limited, mathematical analysis of the behavior of neural networks in “general” cases is typically hard. Similarly to many theoretical works, we focus on a specific “toy” problem setting to demonstrate the phenomenon we wish to study. For this, we chose to use the Parity problem as a canonical “hard” problem studied in many other works.
>
> 2. We agree with the reviewer that curriculum learning is very related to the topic of our paper. However, curriculum learning studies how the order of learning of examples in a given dataset can improve or speed up learning, while we focus on how examples from one task improve the performance on another task, when both tasks are trained together. We are unaware of datasets that are specifically used for curriculum learning which are directly relevant for us to study. That said, following your suggestion, we will add an experiment on a natural dataset in the final version of our paper.
>
> Please consider raising your score if you feel that your concerns were answered, and if not we are happy to continue discussing your comments.

---

> > ### Comment · Reviewer_JC1K · 2023-11-23
> >
> > I appreciate the authors' responses. I still have concerns about the first point since it is still unclear whether the conclusion of this paper could extend to more general scenarios. Therefore, I will maintain my score.

---

### Official Review · Reviewer_JwBs · 2023-11-01

**Soundness:** 3 good
**Presentation:** 3 good
**Contribution:** 3 good
**Rating:** 6
**Confidence:** 4

**Summary:**

Intermediate supervision in the form of chain of thought reasoning traces has been a huge success in improving the reasoning abilities of large language models. However chain of thought requires using more compute at inference time, since the model has to generate many tokens to produce the chain of thought. This work explores an alternative approach, which involves training models in either the multi-task or multi-label setting. The idea being that training the model to complete simpler tasks can help it to learn to solve more complex tasks. They explore this on a simple parity task, which has been shown to be challenging to neural networks to learn, and on a math reasoning task and a code interpretation task. In each setting they observe that mixing simpler tasks into the dataset or defining a multi-label problem, helps the model to solve the more complex task. Without the additional supervision, the model either fails to learn at all or learns much more slowly. They also provide some theoretical results on their parity task. The results pain an interesting picture about why language model pretraining is so effective.

**Strengths:**

* They include both interesting theoretical and convincing empirical results
* Many of their tasks show a dramatic difference between with/without additional supervision, potentially making these tasks a good source for future work to further study
* Their results seem to hold across several tasks and on full transformer language models
* The paper is overall well presented

**Weaknesses:**

* The tasks they study are a little bit toy, which makes them easy to study, but it is a little bit unclear if these findings transfer cleanly to the LM pretraining setting.
* They frame this as a replacement for chain of thought, and state that it saves on collecting full reasoning chains for supervision. However chain of thought prompting (the predominate way to get LM to output intermediate reasoning) only requires a handful of examples that can usually be written by a single person in a few minutes. Whereas their method would require collecting or synthesizing large datasets. I feel that a potentially more interesting framing could be around understanding how large scale pretraining (multitask) can enable LMs to learn tasks which are typically challenging for neural networks to learn on their own.

**Questions:**

* Hoes does model scale impact the results? Will larger models be able to learn more effectively from fewer examples of the simpler tasks?
* Are your models initialized from scratch or fine-tuned from pretrained weights?

---

> ### Author Response · Authors · 2023-11-20
>
> We would like to thank the reviewer for their time reviewing our work. We address the weaknesses below.
>
> **Choice of Synthetic Datasets:**
>
> In our study, we opted for synthetic datasets primarily due to their simplicity and clarity in defining tasks. While we recognize that this choice might limit the direct applicability of our findings to more complex, real-world scenarios, we believed it was necessary for examining the contribution of implicit intermediate supervision in a methodical way.
>
> We acknowledge the importance of demonstrating that our findings are applicable in real-world LM pretraining settings. To this end, we refer to the recent model, OpenHermes-2.5-Mistral-7B, a state-of-the-art 7B parameter Language Model. During its development, the authors found that mixing a natural language instructions dataset with a code instruction dataset in a good ratio (around 10% of the total dataset) boosts model performance on several non-code benchmarks, including TruthfulQA, AGIEval, and GPT4All suite [1]. This aligns directly with our findings, suggesting that our conclusions are not only theoretically sound but also practically applicable in advanced LLM pre training scenarios.
>
> Further substantiating our findings, existing academic literature, such as [2], also aligns with our conclusions. While the focus of [2] is primarily on the impact of pretraining with code to enhance reasoning capabilities in LLMs, it inadvertently supports our theory regarding implicit intermediate supervision. It is important to note that our work complements these existing studies by providing a theoretical underpinning for the observed empirical benefits, thereby filling a crucial gap in the current understanding of LLM training methodologies.
>
> **Framing:**
>
> We agree with the reviewer that the primary contribution of our work lies in settings like pre-training, where intermediate supervision is implicit and does not require the collection or synthesis of additional datasets.
>
> **Questions:**
>
> 1. We agree with the reviewer that the question of how scaling affects implicit intermediate supervision is an interesting question that should be addressed in future work. Note that both our theory and the conducted experiments are in the (relevant) overparameterized regime
> 2. To minimize the potential for unrelated distractions, we initialized the models from scratch.
>
> [1] https://huggingface.co/teknium/OpenHermes-2.5-Mistral-7B
>
> [2] https://aclanthology.org/2022.emnlp-main.90.pdf

---

> > ### Comment · Reviewer_JwBs · 2023-11-20
> >
> > I appreciate your response. I will keep my score the same. Though, it would be great to see the paper more clearly reframed around how your work applies to pre-training, rather than the chain-of-thought angle that the paper currently takes. I agree that there is at least some existence proof for real-world phenomena similar to those you study in this work, and potentially making this clear in the paper would be very helpful.

---

### Meta-Review · Area_Chair_EaHT · 2023-12-09

**Metareview:**

This paper shows that training models on easier multi-task or multi-label setups help with solving more complex tasks, and views it as a way to provide “implicit intermediate supervision”. That is, without an explicit approach like chain-of-thought, the model can learn from easier intermediate steps and solve a more complex task that it would otherwise fail at or learn more slowly. They provide more in-depth theoretical analysis of a simple parity task with a feed-forward architecture. They also show experimental results generalizing to transformer models on other datasets, e.g. math and code interpretation.

Overall, the reviewers have found the theoretical and empirical results interesting. The paper is well-written, and it could potentially have insights for better efficient training schemas.

However, there are major flaws with the current framing. The authors motivate this work by comparing it to CoT, and claiming that it is more challenging. It is correct that the inference is slower with CoT, but providing a few hand-crafted examples is not more challenging than curating a big dataset to train on. In addition, this motivation does not play any role anywhere else in the paper. For example, there is no comparison to CoT, either empirically or theoretically or even on a conceptual level. The parity task is a constrained synthetic setup, and some of the assumptions made are too strong, as highlighted by one of the reviews.

During the discussion period, authors have claimed that relaxing some assumptions would still work, and that they would update the framing, as well as making other adjustments to address other reviewer comments. But there are no changes made to the draft. Therefore, I think the paper would benefit from another round of revision before publication.

**Justification For Why Not Higher Score:**

- The motivation and framing are disconnected from what the paper actually presents.
- The presentation can be significantly improved by incorporating the reviewers' suggestions.

**Justification For Why Not Lower Score:**

N/A

---

### Decision · Program_Chairs · 2024-01-16

Reject